# Refugee Camps as Torturing Environments—An Analysis of the Conditions in the Moria Reception Center (Greece) Based on the Torturing Environment Scale

**DOI:** 10.3390/ijerph191610233

**Published:** 2022-08-17

**Authors:** Pau Pérez-Sales, Andrea Galán-Santamarina, María Victoria Zunzunegui, Sara López-Martin

**Affiliations:** SiR[a] Center and Community Action Group, 28029 Madrid, Spain

**Keywords:** Torturing Environment Scale, Moria camp, Ill treatment, refugees

## Abstract

Background. European countries apply a policy of deterrence of migrants in territorial and extraterritorial border areas. The authors apply the model of torturing environments, which has been already applied to other contexts where persons are deprived of liberty, to the situation of the reception center of Moria, on the island of Lesvos (Greece). Methods. A cross-sectional study was conducted in the months of April and June of 2020. Personal interviews were conducted with 160 people (80 men, 80 women) from Afghan, Syrian, Somalian, and Congo backgrounds. The authors applied the Torturing Environmental Scale, which measures interpersonal violence, emotional distress, and legal safeguards. Results. The findings confirm the inhumane living conditions for the people sheltered in Moria, documenting the severe suffering of the population due to elements linked to basic human functions (hunger, thirst, hygiene, overcrowding, temperature, etc.), actions that produce fear and distress, actions that produce helplessness and hopelessness, actions that cause physical pain, attacks on sexual integrity, and attacks on identity and the need to belong. Some of the data suggest that the purposive and intentionality elements of the definition of cruel, inhuman, or degrading treatment were also met. Conclusions. According to the conceptual model of torturing environments, the Moria reception camp constitutes a space of systematic ill treatment that vulnerated the European legal standards related to torture (Article 3 of the Human Rights Convention). The idea of torturing environments provides a valuable avenue to assess human rights violations in collective spaces and could constitute a useful tool in forensic and litigation processes.

## 1. Introduction

A torturing environment is a space in which conditions are created that, taken together, would meet the United Nations legal definition of torture [1]. It is composed of a set of contextual elements and practices that diminish or override the victim’s will and control over his or her life and compromise the self. The concept of torturing environments goes beyond physical torture to consider those elements of psychological torture that are essential to the understanding of torture in the 21st century [2]. Both medical and legal evidence support and give substance to this notion. Besides physical and psychological elements, a torturing environment integrates legal defenselessness, as the Rapporteur against Torture points out in a recent report [3], which contributes powerfully with its arbitrariness to the scenarios of torture.

Imagine a person who receives little food and in poor condition; is subjected to living conditions without privacy or access to information; is separated from her children; is in a context of constant noise, temperature, or humidity that prevent normal sleep; and is subjected to violent and humiliating daily treatment without legal defense. It is in the global consideration of the combined and accumulated effects of these elements where the consideration of a torturing environment lies although none of these conditions separately would be considered an element of torture by a legal actor. The elements of the environment where this person lives create an effect that leads us to qualify it as an environment of ill treatment or torture when the purposive and intentionality elements of the UNCAT definition are also met [1,2,3].

This paper is part of a body of research that attempts to document the situation of places of detention of migrants from the perspective of their potential consideration as environments of ill treatment and torture [2,4,5,6]. Among these investigations are those that have examined the Moria refugee camp on the Greek island of Lesvos opened in 2015 as a consequence of the decision set out in the European Agenda on Migration to establish a First Reception Centre there for identification, determination of nationality, medical examination, and information on international protection [7]. This refugee camp was closed in September 2020 as a result of arson by its inhabitants in repudiation of the continuous human rights violations that took place there.

This paper aims to analyze the evidence that can support the consideration of the refugee camp of Moria (Lesvos) as a torturing environment.

## 2. Materials and Methods

A cross-sectional study was conducted in the months of April and June of 2020. The study population was the refugees residing in Moria during the study period. As noted by Amnesty International, the 3000-capacity Moria camp was housing 20,000 people in March 2020 (of whom between 6000 and 7000 were under the age of 18) [8].

The sample was selected using a stratified random sampling method that is used to ensure representation from all the subgroups that exists in the camp based on age, gender, an ethnic group [9]. Additionally, the population was stratified by geographic area inside the camp to ensure the collection of information about the different situations that could occur depending on the different areas of the camp. In each area, random routes were used, recording gender and groups according to geographic origin and large linguistic groups (two ethnic groups of Afghans, Arabs, Somalis, and French-speaking Africans) to guarantee a representative sampling in relation to the existing population percentages of each nationality in the camp. In addition, a sample size was also sought that complied with minimum values to guarantee sample representativeness (90% confidence level; 5% margin of error).

The data collection was done through questionnaire by personal interview. The questionnaire was translated into English, French, Somali, Arabic, and Persian. Thirteen interviewers were trained, having been selected from the camp population itself, with at least two interviewers per language and culture. A training was conducted concerning the background and aim of the questionnaire as well as the elements to be taken into account when applying it in the interviews. After a first start, a new space for reflection was generated to assess aspects to be improved and thus unify the methods of intervention, guaranteeing the quality and consistency of the results.

### 2.1. Measures

#### 2.1.1. Torturing Environment

In epidemiological terms, any element of everyday life can be part of a torturing environment if it has been used as a way to provoke or aggravate the physical or psychological suffering of a population [2]. The Torturing Environments Scale (TES) specifically measures the existence of these contexts [10] (Appendix A). It is a scale based on this theoretical model that measures how acts of torture affect a human being. Furthermore, it classifies methods of torture not by how they are carried out but by the impact these methods are intended to have on the person. It includes 8 sections on attacks on basic human functions, actions that produce extreme fear and distress, actions that produce helplessness and hopelessness, actions that produce physical pain, attacks on sexual integrity, and attacks on social identity and sense of belonging.

#### 2.1.2. Interpersonal Violence

Questions on frequent interpersonal violence specific to life in a refugee camp were used according to previous studies.

#### 2.1.3. Emotional Distress

An instrument designed by WHO for use with refugees was used, which asks about severe symptoms of common distress and disturbances in functioning [11].

#### 2.1.4. Legal Safeguards

A list drawn up by jurists was used to determine the degree of compliance with the legal standards that must be guaranteed to the population arriving at a migrant reception center according to the international legal system. It is composed of 20 items distributed in four blocks of rights: those that assist the person concerning the asylum procedure, the right to be informed about the norms and rules of the camp, those related to medical care and associated safeguards, and the right to effective access to a procedure of recourse or legal defense in case of refusal of the asylum procedure.

Information was collected on gender, age, nationality, length of stay in the camp, the existence of a family member in the camp, and information on vulnerability factors before the arrival in Moria: torture, serious physical illness, and mental illness.

### 2.2. Ethical Aspects

The anonymous and voluntary nature of participation was explained, and it was assured that in no case would the information provided have a direct bearing on the individual treatment received in the camp.

Psychosocial support and care for the interviewees and interviewers were provided during the investigation process.

## 3. Results

A total of 160 people were recruited: 80 men and 80 women. Only two people were minors (although close to the age of majority), and three were over 65 years of age. The average age was 30 years. Half (50.1%) were from Afghanistan, followed by Syria (13.8%), Somalia (12.5%), and Congo (10%) (Table 1).

The majority (64.4%) had been in the camp for between 5 and 8 months; 54% had family members in the camp due to the breakdown of the entire family nucleus.

Sixty-five per cent had some kind of vulnerability: a history of torture in their country of origin or during the migratory journey (*n* = 51); had a chronic, serious, or difficult-to-treat illness (*n* = 34); or mental illness, including post-traumatic stress disorder (*n* = 24).

### 3.1. Attacks on Basic Human Functions

Hunger. All (100%) of those interviewed reported having suffered hunger in the camp. Multiple causes are mentioned: quantity (calories, mealtimes), quality (palatability and healthiness), type (specific food, cultural elements), and organizational factors (waiting lines, violence at distribution points, impossibility of conservation).

Thirst. Ninety-seven per cent suffered thirst due to insufficient water for daily consumption. Water for derivative use (washing clothes or other) is not included. The supply of bottled water was restricted in time and quantity and was an instrument of violence and marketing. Access through water points in cleaning places or toilets left doubts about quality and potability.

Bathroom and personal hygiene: difficulties of access, lack of hygiene, and lack of privacy. Everyone reported restricted possibilities for urination, defecation, or personal hygiene and similar problems in being able to wash or shower, with precarious conditions, cold water, and significant risks to the safety of women.

For more than 90%, the camp was an unhealthy space full of garbage, producing or aggravating diseases derived from them. To this, they added the impact of the psychological suffering of having to live in this environment.

Overcrowding was reported by 89% of people, which effectively corresponds to the census population in the camp compared to the maximum declared capacity of the camp. For most people, this means psychological suffering due to the lack of privacy.

Exposure to extreme temperature or humidity conditions. All (100%) of the interviewees expressed having suffered extreme temperature conditions, both cold and heat and humidity conditions, without the possibility of protecting themselves against them.

Temperatures in summer reach 40 °C and in winter can reach 5 °C at night, with moderate rainfall. The building structures were minimal and generally dedicated to the sanitary or service area. Only the area for unaccompanied children had a concrete structure. The central building of the camp was set up around macro-tents without air-conditioning. Some people were in prefabricated constructions and conglomerate spaces without doors.

Most of the population lived in self-constructions made of branches and plastic sheeting supplied by UNHCR with a little more resistance to the weather, but in others, they were thin, plastic sheeting bought from local shops. The absence of electricity made it impossible to have any heating (small stoves) or cooling devices (standing fans or similar).

Sleep problems. Ninety-seven per cent reported problems in getting a minimum of 4–6 h of restful sleep. Noise, fear of aggression and robbery, and temperatures are among the main causes cited.

Health care. Ninety-one per cent stated difficulties in accessing health care within the camp. A small team of Greek doctors provided healthcare, with the presence of international organizations. Care was particularly deficient in minor medical pathology and especially analgesia, limited access to chronic treatments (diabetes, asthma, etc.), emergency care, mental health, and drug addiction.

### 3.2. Actions That Produce Fear and Distress

More than half of those interviewed stated that they had suffered threats that came from conflicts that arose within the camp. Thirty-four per cent stated that these threats were against a family member or acquaintance. Out of those interviewed, 65% reported feeling fear derived from threats, especially threats of aggression against women or children in the context of conflicts between groups of different nationalities. In one-third of these cases, the authorities failed to provide help.

A fourth of the interviewees stated that they had been exposed to environments that could be considered intimidating: fences, barbed wire, concertinas, military checkpoints on access roads, and police and military controls in the interior. From the sample, 78.1% of people claimed to have witnessed violence from other people.

Seventy-nine per cent of the interviewees reported having been robbed by other people sheltering in the camp. Two people reported being robbed by police or camp staff. Further, 80% of the people interviewed reported being afraid or anxious due to the lack of information regarding their administrative situation, their documentation, etc.

### 3.3. Actions That Produce Helplessness and Hopelessness

Sixty per cent of respondents reported the impossibility of establishing routines that give a perception of order or security. They described an arbitrariness and unpredictability of living conditions (meals, sleep cycles, constant changes of lodging place or location in the camp, etc.). Sixty-four per cent stated that they lacked information about the norms and rules of the camp, which generated a perception of defenselessness and vulnerability.

Seventy-five per cent had experienced what they perceived to be lies or deception by officials. This refers to false information, accusations, or evidence regarding transgressions of the law, reasons for arrest or prosecution, and others.

Thirty-four per cent reported not having had an adequate judicial procedure, eight per cent reported not being able to report ill treatment, and six per cent reported having suffered judicial arbitrariness.

### 3.4. Actions That Cause Physical Pain

Thirty-six per cent of people report having been punched, kicked, and/or slapped or having been shaken, dragged, or thrown. More specifically, 20% said they had been whipped or hit with truncheons, batons, or cables that do not leave permanent marks. These aggressions happen mostly among people who live in the countryside although sometimes the police had exercised this violence.

Death was present despite the youthfulness of most of the population. During the last four years of Moria’s operation, 14 deaths were recorded, namely 12 males and 2 females, and 6 were children, including one baby. Five died as a result of fires or explosions in the tents; three from smoke inhalation; three were stabbed in incidents between camp residents; one person committed suicide; one baby died from dehydration; and another child was run over.

A high frequency of intimidation and aggression of an intra-family and intra-community nature was described: 17% (mostly women) reported intra-family or gender-based violence by family members or friends. In 12%, the violence included sexual intimidation (limitations to move freely even for basic needs because of perception of high risk of assault). Sexual exploitation (forced sexual relations in exchange for money, food, or others) was reported by 5%.

### 3.5. Attacks on Sexual Integrity

Seventeen per cent of the respondents stated that they had received humiliation or humiliating treatment of a sexual nature. Of these 31 people, 24 were women. Overall, 17% of the women reported having suffered intra-family or gender-based violence. Nearly half of the women (41%) reported having suffered sexual abuse (e.g., touching, groping, etc., without the use of intimidation and/or violence). Nine per cent of the men interviewed also reported having experienced this type of violence. Within the sample, 12% of the women and 5% of the men interviewed reported having been sexually assaulted (sexual attacks that had applied the use of intimidation and/or violence). Furthermore, 19% of the women suffered either threats or the dissemination of photographs or recordings of a sexual nature; 5% reported having been subjected to situations of sexual exploitation (abuse of someone’s position of vulnerability, differential power, or trust to obtain sexual favors, including but not only by offering money or other social, economic, or political advantages). Moreover, several people had to be evacuated from the camp because of homophobic aggression.

In the Afghan population, virginity tests were conducted to arrange marriages within the camp. In some cases, medical services were asked to carry out such tests. The authorities neither encourage nor prohibit such practices although in many cases, underage girls are involved.

### 3.6. Attacks on Identity and the Need to Belong

Nineteen per cent reported attacks on the self or sense of self, such as being required to break with the past or their previous identity. Illustrations frequently mentioned were harassment or questioning for wearing traditional dress and veiling for impeding worship (10%) and destruction of books or objects of worship predominantly by people of other faiths or opposing religious groups (8%).

Thirty-one per cent considered the camp to be a prolonged environment of submission and inescapable obedience. Another 15% reported cultural isolation, having been forced to live with ethnic, cultural, or religious groups different from their own, which makes communication impossible and leads to the loss of networks of mutual support and a sense of belonging.

Twenty-eight per cent of respondents reported having suffered emotional distress due to the occurrence of intense feelings of guilt. This was due, among other reasons, to the impossibility of caring for dependent family members, protecting family members or friends, or situations of choices in which a third party would be harmed.

Twenty-one per cent reported having suffered induced humiliation and/or embarrassment not related to sexual integrity. Examples include feeling that they are treated like an animal, feeling unworthy of being unable to wash themselves or keep their clothes clean, having to perform acts considered intimate or personal in public, humiliating or degrading treatment by people distributing water or food or controlling access, and separating the family and placing them in different parts of the camp or even in different camps without attending to people’s demands in this regard.

Forty-two per cent reported stigmatization by individuals or the state, including public attacks, threats, or vilification through the media or social networks or through community or membership networks. Examples include racist raids by the police at the request of local traders, racism in the city itself or nearby towns, and public statements by local authorities or bodies of a racist or xenophobic nature. Seventy-four per cent reported discrimination or violence based on religious or ethnic group affiliation. One-fourth reported having suffered this type of discrimination by the police or camp staff.

Figure 1 summarizes the above findings:

### 3.7. Impact on Mental Health

Almost the whole sample reported feeling fear, of which almost one-third reported extreme, intense, or prolonged fear (Table 2). Ninety-eight per cent reported feeling anger, and of these, 31% reported it as extreme, intense, or prolonged. Apathy and loss of interest appeared in 96% of those interviewed and hopelessness in 89%. In its extreme or prolonged form, loss of interest appeared in 35% of people and hopelessness in 31%. Both are indicators of depressive conditions.

Ninety-four per cent of those interviewed have had suicidal thoughts. Of these, 30% had these ideas continuously or repeatedly and the rest occasionally.

All these emotions and symptoms have a functional impact: ninety-nine per cent of respondents reported difficulties in carrying out activities of daily living, and this difficulty was extreme in 35% of them.

### 3.8. Legal Safeguards

Eighty-one per cent reported not having received any information on arrival about the asylum procedures. More than half said that they did not receive legal advice and the available options. Sixty-five per cent reported not having received information about the asylum interview from any official service, and 40% reported a lack of information about their right to have the asylum interview in the presence of a lawyer. Even among those who were assisted by a lawyer at the interview, 60% reported not having had access to all the documentation in their file.

The right to have an interpreter during the asylum interview was often violated. Despite the effective presence of interpreters, more than half of the people surveyed indicated difficulties in guaranteeing the effective transmission of their history to the files. Thus, 58% stated that the interpreter assigned to them was unaware of their context of origin and could not take their cultural codes into account when translating their statements, and 40% reported interference in the interpretation due to pressure or prejudice related to gender, sexual orientation, or ethnic group on the part of the person in charge of translating. Fifty-nine per cent reported not having access to the translation of the documents they had to sign at the asylum interview, and 63% reported not having access to the translation of the asylum decision either.

Forty-seven per cent indicated that they had no information about their rights and duties before they arrived at the camp, 76% did not have this information at the time of their arrival at the camp, and 64% held the official services responsible for this lack of information.

Seventy-two per cent stated that they had no information about the clinical examination before the asylum interview, and 67% had not received any information about its examination. Fifty-five per cent reported that their questions about the examination were not answered during or after the examination. Only half of the people reported having had these medical examinations in conditions of privacy and confidentiality.

Sixty per cent of the respondents reported a lack of information and facilities for effective access to legal assistance—individual or ex officio—in the case of rejection of the asylum procedure. Among those rejected, 59% stated that they have not had legal assistance to file an appeal; 38% emphasized the factual impossibility to file such an appeal, and 61% reported a lack of freedom or obstacles to hiring the services of a private lawyer.

## 4. Discussion

The findings confirm the inhumane living conditions for the people sheltered in Moria. In the months before the fire, conditions were even worse than they had been since 2015 [12,13,14,15,16,17,18,19,20].

The results indicate worsening living conditions, which were already extreme levels of insecurity, fear, and violence slightly higher than those found in previous years in the field [12,17] and a higher frequency of sexual violence [17].

Deterioration of mental health was also evident: high depressive symptomatology in most of the population and a high prevalence of suicidal ideation were found at levels similar to or slightly higher than those found in previous studies [12,21,22].

The main element that stands out in the creation of a torturing environment is the contextual living conditions: difficulties in accessing food, water, overcrowding, unsanitary conditions, temperature and humidity, difficulties in sleeping, and difficulties in receiving health care make up a space for daily life that in itself already constitutes a torturing environment. The second element that stands out is the creation of an environment in which fear and absence of control over one’s own life were continuous given the omnipresence of direct physical violence, witnessing violence against others, and theft and threats. Lastly, legal defenselessness was demonstrated as the absence of administrative or legal information and the anguish generated by the lack of definition and the absence of a future contributed to fear and lack of control. Sometimes, this is added to ambiguous, coercive, or manipulative interrogations in the framework of interviews during the process of applying for international protection.

Within this violence, it is worth highlighting the high level of exposure to sexual violence, especially against women. In addition to abuse and rape, humiliation and humiliation related to access to the bathroom, menstruation or toilet, lack of privacy, and other gender-related elements are widespread. Although most of the reported violence came from situations among the housed persons themselves and not by public officials, these acts of violence derived directly from the conditions of overcrowding, the environment and the hopelessness it generated, the situation of fear, the surrounding chaos, and the impunity towards the perpetrators of the acts of violence.

The breakdown of community ties and family and cultural isolation with the loss of elements of collective identity that allow forms of coping and mutual support contributed to the suffering.

Finally, although it is not a necessary criterion for the definition of a torturing environment, we will argue that there was intent to cause the suffering described here on the part of the European Union authorities and the Greek government for the following reasons:

(a) The coexistence of a physical and living environment full of deprivation, violence, fear, and attacks on individual and collective identity was not an isolated event but points to a political will that sustains it and makes it possible.

(b) Cases of denial of assistance by the camp authorities or the police in contexts of violence and violation of rights had been noted.

(c) Reports of the seriousness of the situation were available to the Greek government and the European authorities for at least eight years without corrective action being taken.

(d) Reports demonstrated the high level of physical and psychological suffering of the persons housed there without any measures being taken to alleviate it.

(e) The prolongation and persistence over time of these acts support intentionality.

(f) It is possible to delimit a combination of purposes that give meaning to the existence of the camp as an environment of torture. On the one hand, it is a punitive action against the persons housed; on the other hand, this context has a dissuasive effect on other persons at risk or under threat in their country of origin who could consider this as a way of fleeing to seek international protection.

In summary, the Moria camp was a space that, from a medical–psychological analysis, constituted a torturing environment. Its central purpose was to serve as a deterrent example for future migrants to Europe so that people fleeing contexts of poverty or violence would be inclined to continue in the situation they were fleeing to avoid the suffering caused by the supposedly receiving European society.

## 5. Conclusions

This paper analyses the Moria camp, which was operating on the island of Lesvos between 2015 and 2020, showing this space as an example of a torturing environment in the framework of the reception centers for migrants generated by European migration policies.

The concept of torturing environments captures the reality of those spaces (concentration camps, detention centers, etc.) that, by their nature, intentionally provoke severe physical or psychological suffering with attacks on the basic and higher functions of human beings by state actors or those with delegated state functions. The central purpose, in the case of centers for migrants or refugees, is to apply policies of a dissuasive nature that generate in people fleeing contexts of poverty or violence the dilemma of having to choose between the situation from which they are fleeing or the suffering caused by the supposedly receiving society as an instrument of control.

This concept goes beyond the classical conceptualization of torture as the infliction of physical pain to break a person’s will but urges to consider from a holistic and integral vision of the human being all the elements of psychological torture that current science shows as essential elements for the understanding of torture in the 21st century.

As a result of these findings, there is a need to create a jurisprudential body that protects and supports the notion of environments of torture and that constitutes elements of enforceability of rights against the states of the citizens subjected to them.

Finally, it is extremely significant that these events take place on European territory in the framework of consolidated democracies in which the right to live in an environment free from torture and under the protection of the state should be an inalienable element.

## Figures and Tables

**Figure 1 ijerph-19-10233-f001:**
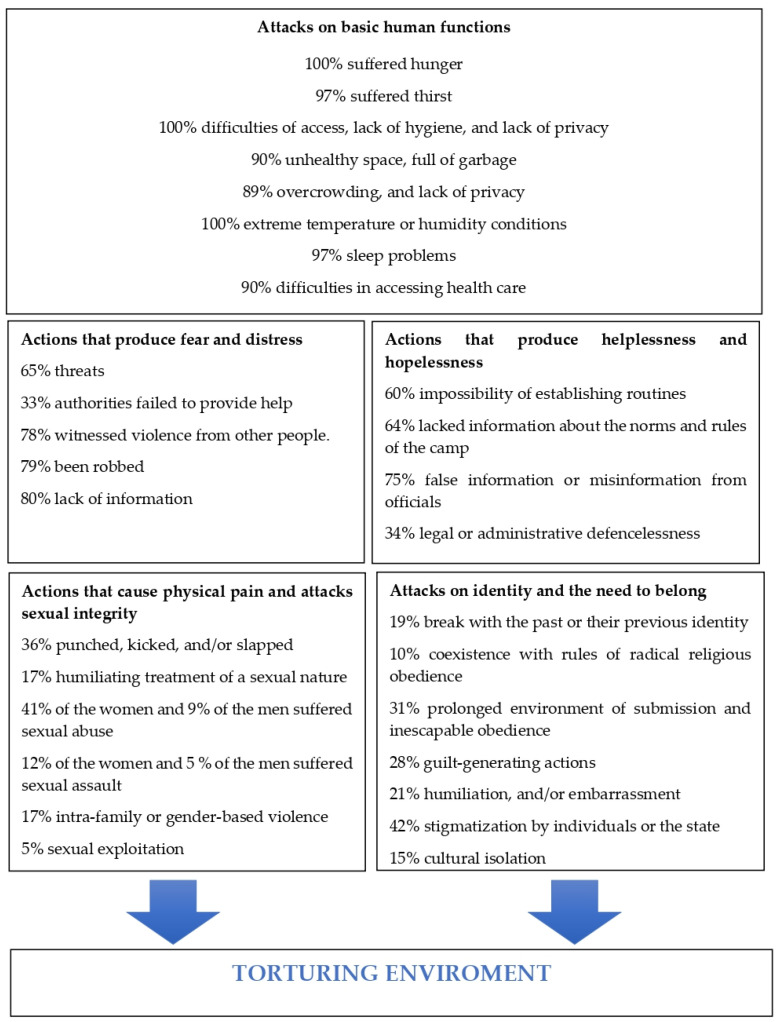
Main findings of the research.

**Table 1 ijerph-19-10233-t001:** Sociodemographic data of the sample.

Variable		N (%)
Gender	Women	80 (50)
	Men	80 (50)
Age	<18	2 (1.3)
	18–65	155 (96.8)
	>66	3 (1.9)
Nationality	Afghanistan	80 (50)
	Cameroon	8 (5)
	Congo	16 (10)
	Ivory Coast	3 (1.9)
	Guinea	1 (0.6)
	Iraq	7 (4.3)
	Mali	2 (1.3)
	Palestine	1 (0.6)
	Syria	22(13.8)
	Somalia	20 (12.5)
Risk of vulnerability		104 (65)
Time in the camp	0–4 months	21 (13.2)
	5–8 months	103 (64.4)
	9–12	35 (21.8)
	>1 year	1 (0.6)
With family members in the camp		87 (54.4)

**Table 2 ijerph-19-10233-t002:** Mental health impacts on people in the Moria camp.

Mental Health Impact	Total (%)	Extremely (%)
Fear	99	32
Anger	97	31
Apathy	96	35
Hopelessness	89	31
Suicidal ideation	94	29
Functional impact	99	35

## Data Availability

Database can be shared on request.

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
