# Peer review of "Refugee Camps as Torturing Environments—An Analysis of the Conditions in the Moria Reception Center (Greece) Based on the Torturing Environment Scale"

_ijerph, 2022, doi:10.3390/ijerph191610233_

Round 1

Reviewer 1 Report

This is a fascinating essay on an immensely important and timely topic.  Although the research and methodology are sound, a bit more detail would bolster the contribution.  In particular, the authors might provide more information about the representativeness of the sources.  Were the respondents a representative cross-section of the camp population?  Did the use of a questionnaire bias the data in any way by excluding non-literate refugees? 

I also recommend that the authors provide a fuller exploration of "agency."  That is, who inflicted the torture?  The paper too often employs passive-voice construction, thus obscuring the people or institutions responsibility for the mistreatment and for creating the environment of ill-treatment.  Moreover, was the neglect the product of indifference or was it purposeful?  If the latter, by whom and toward end?

Adding or enhancing these two elements would make an already strong, important paper more effective.

Author Response

This is a fascinating essay on an immensely important and timely topic. Although the research and methodology are sound, a bit more detail would bolster the contribution.

Thank you very much for this initial feedback, which encourages the whole team.

In particular, the authors might provide more information about the representativeness of the sources. Were the respondents a representative cross-section of the camp population? Did the use of a questionnaire bias the data in any way by excluding on-literate refugees?

Thank you for this comment. We have added some information regarding the nationality distribution to clarify that these percentages are due to ensure the representativeness of the sample. In addition, gender, language, and the spatial layout of the camp were also taken into account as measures to ensure representativeness. The information was collected through a questionnaire and a personal interview, in which the questionnaire was always filled in with the support of a trained interviewer (more information on this is also added in the paper). In this sense, it is not necessary to be able to read and write in order to fill in the questionnaire in any case

I also recommend that the authors provide a fuller exploration of “agency." That is, who inflicted the torture? The paper too often employs passive-voice construction, thus obscuring the people or institutions responsibility for the mistreatment and for creating the environment of ill-treatment. Moreover, was the neglect the product of indifference or was it purposeful? If the latter, by whom and toward end?

Thank you for your comment. We have added further information in relation to this comment in the discussion and conclusion.

Adding or enhancing these two elements would make an already strong, important paper more effective.

Reviewer 2 Report

This paper applying the Torturing Environment Scale conceptual frame to the Moria hotspot in the Aegean Sea represents a fresh angle to illustrate the magnitude of human rights violations. This approach has the potential to deliver new scientific results when measuring the sub-standard conditions in such reception facilities. The selected conceptual framework applies an appropriate methodology, although it is suggested that the term “Torturing Environment Scale” be explained already in the Introduction, i.e. what it means and where it originates from.

Even though this is not a legal piece of work, the authors shoud be more cautios when using the terms “torture”, “ill-treatment” and “inhumane treatment” interchangeably. Legally speaking, they refer to different kind of treatment, along a sliding scale of gravity. For instance, the European Court of Human Rights has drawn distinctions between prohibited acts under Article 3 ECHR [this is referred to by the authors], essentially using a “threshold of severity” test. “Inhuman treatment” is defined as at least such treatment as deliberately causes severe suffering, mental or physical, which in the particular situation is unjustifiable. The severity of pain and suffering forms the basis for distinguishing between inhuman and degrading treatment in accordance with ECtHR case law, which establishes that “degrading treatment” must include at least some form of gross humiliation.

Overall, this well-founded and evidence-backed piece gives a new twist to assess the multiple and grave human rights violations in the (ex-)Moria camp. Still, the discussion should be a bit sharper on how all the findings and interview results lead to the conclusion that the former Moria camp qualified as a “torturing environment”. This by better connecting the dots between the findings and the overall assessment based on the methodology/conceptual framework applied. The conclusions are also a bit short and underdeveloped, which needs some beefing up – e.g by elaborating  more on the very interesting last sentence of the abstract.

Further, more specific and detailed comments are included in the attached manuscript.

Author Response

This paper applying the Torturing Environment Scale conceptual frame to the Moria hotspot in the Aegean Sea represents a fresh angle to illustrate the magnitude of human rights violations. This approach has the potential to deliver new scientific results when measuring the sub-standard conditions in such reception facilities.

We are grateful for this positive feedback, which encourages the team to continue working on the same conceptual framework.

The selected conceptual framework applies an appropriate methodology, although it is suggested that the term “Torturing Environment Scale” be explained already in the Introduction, i.e. what it means and where it originates from.

Thanks for the comment. We add some more information as well as the scale validation article is cited so that more information can be consulted.

Even though this is not a legal piece of work, the authors should be more cautions when using the terms “torture”, “ill-treatment “and “inhumane treatment” interchangeably. Legally speaking, they refer to different kind of treatment, along a sliding scale of gravity. For instance, the European Court of Human Rights has drawn distinctions between prohibited acts under Article 3 ECHR [this is referred to by the authors], essentially using a “thresholder severity” test. “Inhuman treatment” is defined as at least such treatment as deliberately causes severe suffering, mental or physical, which in the particular situation is unjustifiable. The severity of pain and suffering forms the basis for distinguishing between inhuman and degrading treatment in accordance with ECtHR case law, which establishes that “degrading treatment” must include at least some form of gross humiliation.

We appreciate the comments on the jurisprudential distinctions between torture, inhuman and degrading treatment, which are obviously not unfamiliar to us. It was not our intention to ignore the " severity " factor, the importance of which is well known to us, when dealing with these terms.

However, our starting point, made explicit in the first paragraph of the text, is that used by the Special Rapporteur against Torture in his report on Psychological Torture (A/HRC/43/49), in which he interprets the concept of "torturing environments" as a set of elements that, on their own, would not reach the necessary severity to be considered "torture", but which are so as a whole, so that it can be understood that they will reach different severity when they operate together, separately or in different combinations. Added to this is the fact that the leap between the different concepts also depends, the Special Rapporteur points out, on the intensity of the cumulative factor:

  1. “68. The above outline of specific methods should not obscure the fact that, in practice, torture victims are almost always exposed to a combination of methods, techniques and circumstances deliberately designed to inflict both mental and physical pain or suffering. If applied in isolation or for a short period of time, some of these techniques and circumstances may not necessarily amount to torture. In combination and with increasing duration, however, they have a devastating effect. Thus, a finding of torture may depend not only on the specific characteristics of particular techniques or circumstances, but also on their cumulative and/or prolonged effect, sometimes in conjunction with external stress factors or individual vulnerabilities that are not under the control of the torturer and may not even be consciously instrumentalized by him. As aptly stated by the International Tribunal for the Former Yugoslavia, torture “may be committed in one single act or can result from a combination or accumulation of several acts, which, taken individually and out of context, may seem harmless ... The period of time, the repetition and various forms of mistreatment and severity should be assessed as a whole”.

Our intention was precisely not to distinguish between "torture" and other types of treatment, this being the feature that we intended to define the text, insofar as we suggest that there may be intentional elements of defenselessness of the victims, of greater or lesser severity, but which together could constitute different types of treatment: degrading treatment, inhuman treatment or torture. In any case, it is not the function of an expert to legally qualify the facts, so we only suggest that in "torturing environments" there could be a violation of art. 3 of the European Convention on Human Rights, it being the function of the judge to determine the type of treatment that specifically applies.

Overall, this well-founded and evidence-backed piece gives a new twist to assess the multiple and grave human rights violations in the (ex-)Moria camp. Still, the discussion should be a bit sharper on how all the findings and interview results lead to the conclusion that the former Moria camp qualified as a “torturing environment”. This by better connecting the dots between the findings and the overall assessment based on the methodology/conceptual framework applied.

Thank you for the comment. We have improved and expanded the discussion in order to better clarify this part of the article.

The conclusions are also a bit short and underdeveloped, which needs some beefing up – e.g by elaborating more on the very interesting last sentence of the abstract.

Thank you for the comment. We have also improved and expanded the conclusions in order to better clarify this part of the article.

Further, more specific and detailed comments are included in the attached manuscript.

Regarding this part, we have already introduced the comments that were done along the manuscript. We attached here the answer to some questions that were made but we considered not to include the explanation in the paper:

  • Was that intentional to cover only two children?

The proposed sample was made up of adults. However, at the time of the research some cases were persons who were classified as adults in the camp but who reported that they were not yet of legal majority and had been classified as such due to the lack of documents. In both cases we are talking about people who were between 16 and 17 years old. This characteristic is mentioned in the paper in order to provide more information on the composition of the sample.

Reviewer 3 Report

An excellent paper worth publishing with some attention to some minor issues.

Section 2. Materials and Methods: Authors need to explain how the sample was chosen. Was it random, or targeted? It is clearly targeted in relation to the gender of respondents (50/50). Why only gender compared with other demographic variables? To what extent is the sample representative of the camp population? This is a more serious issue than the others identified here.

Table 1. Needs some English corrections such as Nationality, Cameroon, Palestine, Syria...

3.1 Further details about "drug addiction" seem required. What kind of drugs, how accessed, etc.

3.2 "conflicts between groups" - what groups specifically and for what reasons. There is some further reference later, but still not enough details. Inter- and intra- group conflicts are important to understand. And it may be too easy to blame all such conflicts (as well as some of the sexual violations) on overcrowding.

3.4-3.5. The categories of sexual violations are confusing. There is sexual: exploitation, humiliation, identity, integrity, orientation, abuse, assault, aggression, threats, harassment (?). Each have different incidence figures. The authors should define each of these classifications more clearly in terms of specific behaviors and perhaps reduce the number of classifications, if possible.

Conclusion: the phrase "...inclined to continue in the situation they were fleeing..." is slightly ambiguous. A clearer statement could be made here. The authors may also add: "or to seek an alternative asylum destination..."

Finally, the riots and closure of the refugee camp came only a few months after the fieldwork for this study. Was there any relationship between the two events? Some comment on this may be useful.

Author Response

An excellent paper worth publishing with some attention to some minor issues.

We greatly appreciate the initial feedback that is important for the entire team thanks to the recognition it entails.

Section 2. Materials and Methods: Authors need to explain how the sample was chosen. Was it random, or targeted? It is clearly targeted in relation to the gender of respondents (50/50). Why only gender compared with other demographic variables? To what extent is the sample representative of the camp population? This is a more serious issue than the others identified here.

Thank you for this observation. In relation to the comment, we have added information on the distribution of nationalities to clarify that these percentages are due to the representativeness of the sample. In addition to gender and nationality as variables to ensure the representativeness of the sample, the spatial distribution of the camp was also considered to collect data from all areas of the camp. In addition, the questionnaires were translated into different languages and the interviewers spoke different languages and belonged to different cultural backgrounds in order to guarantee a good accompaniment during the questionnaire and interview and thus access to people from different socio-educational levels.

Table 1. Needs some English corrections such as Nationality, Cameroon, Palestine, Syria...

Thank you very much for pointing this out. We have already corrected.

3.1 Further details about "drug addiction" seem required. What kind of drugs, how accessed, etc.

Thank you for the observation. Some people pointed out the situation of addiction to substances such as alcohol or glue. Being an element to take into account the consumption of drugs, it was not a central element of the investigation and that is why no further information was included or an in-depth study of the subject was carried out.

3.2 "conflicts between groups" - what groups specifically and for what reasons. There is some further reference later, but still not enough details. Inter- and intra- group conflicts are important to understand. And it may be too easy to blame all such conflicts (as well as some of the sexual violations) on overcrowding.

In several parts of the text the variables that imply inter- and intra- group differences are pointed out. These variables are mainly religion, nationality, and ethnicity. In the text we do not wish to specifically point out any of these variables in an identifying manner as we understand the focus of the article is to identify the torturing environment as the element that promotes violence, and not the characteristics of the people who constitute the groups. On the other hand, in no case do we point to overcrowding as the only factor, but on the contrary the article tries to describe how multiple elements function on the basis of accumulation as a torturing environment as a whole. Among these elements is overcrowding, but not exclusively.

3.4-3.5. The categories of sexual violations are confusing. There is sexual: exploitation, humiliation, identity, integrity, orientation, abuse, assault, aggression, threats, harassment (?). Each have different incidence figures. The authors should define each of these classifications more clearly in terms of specific behavior sand perhaps reduce the number of classifications, if possible.

Thank you very much for your comment, so we have improved the wording of the paragraph on attacks on sexual integrity and added necessary definitions to clarify the information.

Conclusion: the phrase "...inclined to continue in the situation they were fleeing..." is slightly ambiguous. A clearer statement could be made here. The authors may also add: "or to seek an alternative asylum destination..."

We have improved and expanded the conclusions in order to better clarify this part of the article.

Finally, the riots and closure of the refugee camp came only a few months after the fieldwork for this study. Was there any relationship between the two events? Some comment on this may be useful.

We understand that the question has to do with whether the investigation could have had an influence on the riots and the camp fire. We consider that there is no connection, but on the contrary the report helps to understand the context in which this tragic situation ends up happening. This element is a valuable part of the article.

Round 2

Reviewer 1 Report

The revisions improve the paper but do not directly engage the concerns that I raised.  The author(s) has added language about testing for representativeness, but he/she/they does not demonstrate that the respondents to the questionnaire were indeed representative.  This is easily corrected and would strengthen the submission.

Perhaps more important, the author(s) could engage the "agency" issue more effectively.  The revised version notes the design of the camp but does not discuss the purpose of the mistreatment.  Nor does the essay discuss the perpetrators of the mistreatment.  Learning the larger purpose of the institution is valuable but does not engage my concern about "agency."  

In short, the revisions improve the essay, but additional revisions would make this fine paper even more important.

Author Response

The revisions improve the paper but do not directly engage the concerns that I raised.  The author(s) has added language about testing for representativeness, but he/she/they does not demonstrate that the respondents to the questionnaire were indeed representative.  This is easily corrected and would strengthen the submission.

We are grateful for the comment for which we have added some more information.

Perhaps more important, the author(s) could engage the "agency" issue more effectively.  The revised version notes the design of the camp but does not discuss the purpose of the mistreatment.  Nor does the essay discuss the perpetrators of the mistreatment.  Learning the larger purpose of the institution is valuable but does not engage my concern about "agency."  

In relation to this section, we consider that the purpose of the violence described in this article is made explicit, as well as the agents ultimately responsible between lines 352 and 370. We would be grateful if you could give us a more concrete comment in order to improve this part.  

In short, the revisions improve the essay, but additional revisions would make this fine paper even more important.
